# Accuracy Bounds and Measurements of a Contactless Permittivity Sensor for Gases Using Synchronized Low-Cost mm-Wave Frequency Modulated Continuous Wave Radar Transceivers

**DOI:** 10.3390/s19153351

**Published:** 2019-07-31

**Authors:** Andreas Och, Jochen O. Schrattenecker, Stefan Schuster, Patrick A. Hölzl, Philipp F. Freidl, Stefan Scheiblhofer, Dominik Zankl, Robert Weigel

**Affiliations:** 1DICE GmbH & Co. KG, 4040 Linz, Austria; 2FAU Erlangen-Nuremberg, 91058 Erlangen, Germany; 3Voestalpine Stahl GmbH, 4020 Linz, Austria; 4Infineon Technologies Austria AG, 8020 Graz, Austria

**Keywords:** Cramér-Rao bounds, gas detectors, millimeter wave radar, permittivity, phase estimation, radar measurements, time of arrival estimation

## Abstract

A primary concern in a multitude of industrial processes is the precise monitoring of gaseous substances to ensure proper operating conditions. However, many traditional technologies are not suitable for operation under harsh environmental conditions. Radar-based time-of-flight permittivity measurements have been proposed as alternative but suffer from high cost and limited accuracy in highly cluttered industrial plants. This paper examines the performance limits of low-cost frequency-modulated continuous-wave (FMCW) radar sensors for permittivity measurements. First, the accuracy limits are investigated theoretically and the Cramér-Rao lower bounds for time-of-flight based permittivity and concentration measurements are derived. In addition, Monte-Carlo simulations are carried out to validate the analytical solutions. The capabilities of the measurement concept are then demonstrated with different binary gas mixtures of Helium and Carbon Dioxide in air. A low-cost time-of-flight sensor based on two synchronized fully-integrated millimeter-wave (MMW) radar transceivers is developed and evaluated. A method to compensate systematic deviations caused by the measurement setup is proposed and implemented. The theoretical discussion underlines the necessity of exploiting the information contained in the signal phase to achieve the desired accuracy. Results of various permittivity and gas concentration measurements are in good accordance to reference sensors and measurements with a commercial vector network analyzer (VNA). In conclusion, the proposed radar-based low-cost sensor solution shows promising performance for the intended use in demanding industrial applications.

## 1. Introduction

On-line gas monitoring is an increasingly important element in a multitude of industrial applications, for example, in the process [1] or petrochemical industry [2,3], for food processing [4] as well as for waste treatment plants [5,6]. To ensure the proper conditions for both optimal quality and maximum efficiency and furthermore to guarantee a safe operation multiple parameters such as pressure, temperature or gas concentration have to be measured continuously. Therefore, various gas sensing technologies, for example, metal-oxide (MOX) semiconductor, calorimetric and optical sensors, for instance nondispersive infrared (NDIR), are widely used [7,8]. However, invasive sensor technologies such as MOX and calorimetric detectors require direct contact to the medium-under-test (MUT). Thus, they might not be applicable for harsh industrial environments containing hot or corrosive gases. In contrast, NDIR optical sensing is a potentially contactless technique with real-time capability but detects only infrared-active gases [9,10].

A radar based time-of-flight (TOF) sensor for pressure and concentration measurements was first proposed as an alternative gas monitoring solution in Reference [11]. However, it relies on an experimental and relatively large ultra-wideband (UWB) radar. To overcome these restrictions, we introduced a low-cost permittivity sensor using a fully-integrated millimeter-wave (mmW) radar transceiver in Reference [12] which reduces system cost and complexity significantly. In Reference [13] the measurement concept was expanded from reflection- to transmission-based TOF sensing to increase the system’s robustness and accuracy.

This paper focuses on the theoretical and practical performance limits of low-cost frequency-modulated continuous-wave (FMCW) radar sensors for monitoring binary gas mixtures. First, the Cramér-Rao lower bound (CRLB) is derived for relative permittivity and concentration measurements and validated by Monte-Carlo simulations. Based on this, a sensor concept is presented and prototype hardware using fully-integrated transceivers (MMICs) in the frequency range of 76–81 GHz is developed accordingly. Measurements of varying Helium and Carbon Dioxide levels in air are conducted with a distributed sensor configuration using two opposing synchronized radar systems. A compensation method for systematic setup induced errors is introduced and its effectiveness tested. The results are compared to both values obtained with a commercial vector network analyzer (VNA) as well as to an estimate of the ground-truth acquired by traditional MOX gas sensors. Although the number of measurement repetitions is not sufficient to determine the prototype system’s accuracy limit with high confidence the practical variance can be concluded to be in the order of the theoretical limit given by the CRLB. Overall, the proposed sensor concept using synchronized low-cost radar MMICs demonstrates promising performance for demanding industrial applications.

## 2. Materials and Methods

### 2.1. Fundamentals

A sampled sinusoidal baseband (BB) signal received by a FMCW radar sensor can be modeled by
(1)$$x\left[n\right]=A\cos\left(2\pi \psi n+\varphi \right)+\omega \left[n\right]$$
with amplitude A>0, normalized frequency $\psi =f/f_{s}\in \left(0,1/2\right)$ with sampling frequency $f_{s}$ and phase $\varphi \in \left(-\pi ,\pi \right]$. Additive white Gaussian noise $\omega \;\left[n\right]∼N\left(0,\sigma^{2}\right)$ with variance $\sigma^{2}$ represents the unavoidable measurement noise. Time index *n* is in the range of n=0,1,⋯N−1 with *N* being the number of acquired samples. For (Equation 1) the CRLB for normalized frequency and phase are approximately given by Reference [14] as
(2)$$\mathrm{var}\left\{\hat{\psi }\right\}\geq \frac{12}{{\left(2\pi \right)}^{2}\cdot \mathrm{SNR}\cdot N^{3}}$$
(3)$$\mathrm{var}\left\{\hat{\varphi }\right\}\geq \frac{4}{\mathrm{SNR}\cdot N}$$
assuming N≫1 and 0<ψ<1/2. In (Equation 2) and (Equation 3) $\hat{\psi }$ denotes the estimated quantity, SNR is the signal-to-noise ratio $\mathrm{SNR}=A^{2}/2\sigma^{2}$. Information about the TOF $\tau_{d}$ the transmitted radar signal takes to arrive at the receiver is contained in the frequency and phase component of (Equation 1). In case of a FMCW radar system as utilized in this paper, the frequency component
(4)$$f=\frac{B}{T}\tau_{d}$$
is defined by the sweep bandwidth *B* and the sweep time *T* of the FMCW chirp signal. The chirp start frequency $f_{0}$ influences the phase component
(5)$$\varphi =2\pi f_{0}\tau_{d}-\pi \frac{B}{T}\tau_{d}^{2}\approx 2\pi f_{0}\tau_{d}\;.$$

For typical FMCW radar systems, the second phase term in (Equation 5) depending on the chirp slope only has minimal influence on the overall phase offset. As an example, using the parameters of the prototype sensor given in Table 1, $\pi B/T\tau_{d}^{2}\approx 10^{-4}\cdot 2\pi \ll 2\pi f_{0}\tau_{d}\approx 10^{2}\cdot 2\pi$ for a propagation distance of 0.5 m.

The measured TOF
(6)$$\tau_{d}=\tau_{m}+\tau_{\mathrm{ext}}$$
contains not only the time it takes an electromagnetic signal to travel through a medium $\tau_{m}$ but also an additional time delay $\tau_{\mathrm{ext}}$ caused by external signal paths such as the gas container walls, antennas as well as connectors and signal traces on the system’s printed circuit board (PCB). The parameter of interest
(7)$$\tau_{m}=\frac{d_{m}}{c_{0}}\sqrt{\varepsilon_{r}}$$
depends on the material’s dielectric properties. In (Equation 7), $\varepsilon_{r}$ denotes the relative permittivity, $d_{m}$ the physical distance traveled through the medium and $c_{0}$ is the speed of light within vacuum. To eliminate the external influences, the measurement system is calibrated with a known medium, for example, air, with dielectric constant $\varepsilon_{r,\mathrm{ref}}$. Further on, only the relative TOF
(8)$$\Delta \tau_{d}=\tau_{d}-\tau_{d,\mathrm{ref}}=\frac{d_{m}}{c_{0}}\left(\sqrt{\varepsilon_{r}}-\sqrt{\varepsilon_{r,\mathrm{ref}}}\right)$$
is considered. Solving (Equation 8) for relative permittivity $\varepsilon_{r}$, properties of the MUT such as pressure, temperature or humidity can be characterized [15,16,17].

If no pure gas is present, it was shown in Reference [11] that the effective permittivity of a binary gas mixture can be approximated by
(9)$$\varepsilon_{r,\mathrm{eff}}\approx \varepsilon_{r,e}+\frac{3\cdot \zeta_{i}\cdot \varepsilon_{r,e}\cdot \left(\varepsilon_{r,i}-\varepsilon_{r,e}\right)}{\varepsilon_{r,i}+2\cdot \varepsilon_{r,e}-\zeta_{i}\left(\varepsilon_{r,i}-\varepsilon_{r,e}\right)}\;.$$
$\varepsilon_{r,e}$ and $\varepsilon_{r,i}$ are the dielectric constants of environmental and inclusion gas, while $\zeta_{i}\in \left[0,1\right]$ is the volume fraction of the inclusion gas.

### 2.2. Cramér-Rao Lower Bound

To determine the statistically most accurate measurement method given the signal model (Equation 1) the CRLB for TOF estimation is reviewed, once based on frequency *f* and once based on the phase component φ [18,19,20]. Combining (Equation 2) and (Equation 4) yields the CRLB
(10)$$\mathrm{var}{\left\{{\hat{\tau }}_{d}\right\}}_{f}\geq \frac{12}{{\left(2\pi \right)}^{2}\cdot \mathrm{SNR}\cdot B^{2}\cdot N}$$
based on frequency, while inserting (Equation 5) into (Equation 3) gives the phase-based CRLB as
(11)$$\mathrm{var}{\left\{{\hat{\tau }}_{d}\right\}}_{\varphi }\geq \frac{4}{{\left(2\pi \right)}^{2}\cdot \mathrm{SNR}\cdot f_{0}^{2}\cdot N}\;.$$

In most real-world radar measurements, these variances indicate only theoretical limits which cannot be reached due to amplitude modulation effects [21] and a window function applied to the sampled data to reduce the sidelobe level [22,23]. For the sake of simplicity and clarity, both effects are omitted initially but taken into account in the analysis of the Monte-Carlo simulations presented later in this Section. Comparing phase-based to frequency-based variance via the ratio
(12)$$\frac{\mathrm{var}{\left\{{\hat{\tau }}_{d}\right\}}_{\varphi }}{\mathrm{var}{\left\{{\hat{\tau }}_{d}\right\}}_{f}}=\frac{4}{12}\cdot \frac{{\left(2\pi \right)}^{2}\cdot \mathrm{SNR}\cdot B^{2}\cdot N}{{\left(2\pi \right)}^{2}\cdot \mathrm{SNR}\cdot f_{0}^{2}\cdot N}=\frac{1}{3}\cdot \frac{B^{2}}{f_{0}^{2}}\;$$
demonstrates that the achievable accuracy when using phase estimation is far superior to a frequency measurement, as in most practical radar systems $B\ll f_{0}$ (see Section 4, where (Equation 12) is evaluated for the parameters used in the prototype system). While a major drawback of phase estimation is the 2π ambiguity, it was found in Reference [12] that for most common gases the difference in TOF is small enough to stay well within the unambiguous range of ±π. Thus, phase estimation is used as basic principle in this paper. Since the relative TOF is measured after calibration, it follows as worst-case limit for independent measurements with $\mathrm{cov}\left({\hat{\tau }}_{d},{\hat{\tau }}_{d,\mathrm{ref}}\right)\approx 0$:(13)$$\mathrm{var}{\left\{\Delta {\hat{\tau }}_{d}\right\}}_{\varphi }=\mathrm{var}{\left\{{\hat{\tau }}_{d}-{\hat{\tau }}_{d,\mathrm{ref}}\right\}}_{\varphi }\approx 2\cdot \mathrm{var}{\left\{{\hat{\tau }}_{d}\right\}}_{\varphi }$$

Using first-order linear approximation [14] with (Equation 8) as $g\left(\Delta {\hat{\tau }}_{d}\right)$, the CRLB for permittivity calculated from phase-based TOF evaluates to
(14)$$\mathrm{var}{\left\{{\hat{\varepsilon }}_{r}\right\}}_{\varphi }\geq {\left(\frac{\partial g\left(\Delta {\hat{\tau }}_{d}\right)}{\partial \Delta {\hat{\tau }}_{d}}\right)}^{2}\cdot \mathrm{var}{\left\{\Delta {\hat{\tau }}_{d}\right\}}_{\varphi }={\left(2\frac{c_{0}}{d_{m}}\left(\sqrt{\varepsilon_{r,\mathrm{ref}}}+\frac{c_{0}}{d_{m}}\Delta \tau_{d}\right)\right)}^{2}\cdot \mathrm{var}{\left\{\Delta {\hat{\tau }}_{d}\right\}}_{\varphi }\;.$$

For materials with dielectric properties similar to the calibration medium $\varepsilon_{r}\approx \varepsilon_{r,\mathrm{ref}}$ (as it is the case for most common gases),
(15)$$\frac{c_{0}}{d_{m}}\Delta \tau_{d}\ll \sqrt{\varepsilon_{r,\mathrm{ref}}}\;.$$

Thus, (Equation 14) simplifies to
(16)$$\mathrm{var}{\left\{{\hat{\varepsilon }}_{r}\right\}}_{\varphi }⪆{\left(\frac{4\;c_{0}}{2\pi }\right)}^{2}\cdot \frac{\varepsilon_{r,\mathrm{ref}}}{d_{m}^{2}}\cdot \frac{2}{\mathrm{SNR}\cdot f_{0}^{2}\cdot N}\;.$$

As expected, the theoretically achievable measurement accuracy increases with higher SNR, number of samples *N* and chirp start frequency $f_{0}$. Additionally, a longer propagation path $d_{m}$ through the MUT significantly improves the achievable performance. On the other hand, the higher the reference material’s dielectric constant, the higher the variance of the permittivity estimation.

To derive the CRLB for gas concentration, (Equation 9) is first approximated under the assumption $\varepsilon_{r,i}/\varepsilon_{r,e}\approx 1$, which is valid for most common gases. Inserting the simplified solution for effective permittivity
(17)$$\varepsilon_{r,\mathrm{eff}}=\varepsilon_{r,e}+\frac{3\cdot \zeta_{i}\cdot \left(\varepsilon_{r,i}-\varepsilon_{r,e}\right)}{\frac{\varepsilon_{r,i}}{\varepsilon_{r,e}}+2-\zeta_{i}\left(\frac{\varepsilon_{r,i}}{\varepsilon_{r,e}}-1\right)}\approx \varepsilon_{r,e}+\zeta_{i}\left(\varepsilon_{r,i}-\varepsilon_{r,e}\right)$$
into (Equation 16) results in
(18)$$\mathrm{var}{\left\{{\hat{\zeta }}_{i}\right\}}_{\varphi }\approx \frac{1}{{\left(\varepsilon_{r,i}-\varepsilon_{r,e}\right)}^{2}}\cdot \mathrm{var}\left\{{\hat{\varepsilon }}_{r,\mathrm{eff}}\right\}={\left(\frac{4\;c_{0}}{2\pi }\right)}^{2}\cdot \frac{\varepsilon_{r,\mathrm{ref}}}{d_{m}^{2}\cdot {\left(\varepsilon_{r,i}-\varepsilon_{r,e}\right)}^{2}}\cdot \frac{2}{\mathrm{SNR}\cdot f_{0}^{2}\cdot N}\;.$$

Equation (Equation 18) clearly indicates a reduction in variance proportional to the quadratic difference of permittivity of both gases involved. In other words: The closer environmental and inclusion gas match in their dielectric properties, the more difficult it is to differentiate them by a TOF measurement, therefore reducing the accuracy of the volume fraction estimation.

To validate the derived CRLBs as well as to confirm the correctness of the assumptions made to simplify the equations, extensive Monte-Carlo simulations with a minimum of 2000 repetitions were carried out in MATLAB. Data was modeled according to (Equation 1) with a sample length N=4096.

First, a coarse frequency spectrum $X\left(f\right)$ was calculated by means of Fast-Fourier transform (FFT) with low zero-padding factor, followed by two iterative chirp-z transformations resulting in an effective zero-padding factor of $2^{32}$. In accordance to the signal processing implemented for the experimental validation a Hanning window function was applied to the time samples increasing the theoretical CRLB by 3.288dB [23]. Figure 1 and Figure 2 illustrate the inverse quadratic dependency on the length of the propagation path $d_{m}$ and on the chirp start frequency $f_{0}$ for phase-based permittivity measurements by plotting the mean square error (MSE)
(19)$$\mathrm{MSE}\left\{{\hat{\varepsilon }}_{r}\right\}=E\left\{{\left({\hat{\varepsilon }}_{r}-\varepsilon_{r}\right)}^{2}\right\}$$
over the respective sweep parameter. In Figure 3, the CRLB performance limit of an concentration measurement with $\zeta_{i}=0.5$ is depicted for increasing difference of environmental and inclusion gas permittivity.

### 2.3. Measurement Setup

Multiple measurements of CO${}_{2}$ ($\varepsilon_{r,i}=1.000921$) and He ($\varepsilon_{r,i}=1.000065$) as inclusion gases inside air ($\varepsilon_{r,e}=1.000536$) were conducted. For both gases highly accurate data of their dielectric properties is available [15,17,24,25] and they show no significant dependency on frequency in the mmW range [26]. They are easy to obtain and uncritical to handle as they are non-toxic and non-flammable. Two low-cost radar sensors (detailed description in Section 2.4) were placed on opposing sides of the observation area to acquire the TOF through the MUT in a transmission measurement. Figure 4 shows a schematic illustration of the proposed concept.

The binary gas mixture is contained in an acrylic box with dimensions 50×50×30 cm${}^{3}$. It features gas inlets and outlets as well as a removable top lid to access the various sensors mounted inside (see Section 2.5). As the container is not fully sealed, small leakages make gas concentrations over 95% not feasible. A mechanical mounting structure built from Aluminum profile fixes the positions of the radar sensors with respect to the observation area. Adhesive radio frequency (RF) absorbers are used to minimize reflections by the mechanical structure and multi-path propagation inside the gas container. A photograph of the realized setup is depicted in Figure 5.

### 2.4. Radar Prototype Hardware

The low-cost sensor prototype utilizes a fully-integrated FMCW automotive radar MMIC by Infineon Technologies fabricated in Silicon-Germanium technology [27]. All components necessary for radar operation are integrated on chip, such as a FMCW chirp generator for the frequency range 76−81 GHz, the entire transmit chain including power amplifier, a receive chain with digitized output as well as a digital control unit. Out of the three transmitter (TX) and four receiver (RX) channels available, only one channel each is used in this application to form a bi-static radar system. The MMICs are soldered onto development boards featuring 20 dBi standard gain horn antennas for E-band. Evaluation of the measurement data is performed on a host PC controlling the radar sensor via a custom FPGA board. Table 1 summarizes the MMIC configuration parameters used to obtain the results in Section 3.

To enable phase coherent operation of two distributed radar systems, it is necessary to synchronize them with high precision. The MMICs used in the hardware prototype support this operational mode by a master-slave cascading feature, which is usually employed to increase the number of TX and RX channels and thus achieve higher angular resolution [27]. In this configuration the local oscillator (LO) signal of the master MMIC is forwarded to the slave sensor via a WR-12 waveguide of 1.30m total length, the 50 MHz system clock and other digital control signals are synchronized via SMA and signal cables of the same length, respectively. As discussed in Section 2.7, the RF waveguide link is sensitive to environmental temperature changes. This has to be taken into account and corrected in post-processing.

### 2.5. Reference & Ground-Truth Estimation

To obtain reference values for comparison with the low-cost radar hardware, S-Parameter measurements of the same binary gas mixtures were conducted with a Keysight VNA E8361A offering a bandwidth of 67–110 GHz. However, the instrument was equipped with 20 dBi standard gain horn antennas for E-band limiting the available bandwidth to 67–90 GHz. For the reference measurement the radar sensors in the setup shown in Figure 4 are replaced by the mono-static VNA ports connected to antennas TX${}_{1}$ and TX${}_{2}$, while antennas RX${}_{1}$ and RX${}_{2}$ remain unused. All VNA channels are synchronized internally, therefore the synchronization link between the opposing ports can be omitted.

The ground-truth of the volume fraction value is estimated by CO${}_{2}$ and O${}_{2}$ concentration sensors on MOX basis located at the top and bottom of the box. They are specified by the manufacturer to deliver a result accurate to $\pm \left(70\;\mathrm{ppm}+5\%\;\mathrm{of}\;\mathrm{reading}\right)$ and ±2%, respectively, after calibration in air. While CO${}_{2}$ concentration can be measured directly, no economical sensors for He are available and the He fraction is derived from the reduction in O${}_{2}$ content in the air reference. This indirect measurement procedure scales the specified accuracy of the He level reading up to ±10%, though in the experiments much lower variations were observed.

Due to the density difference of environmental and inclusion gas the MOX sensors on top and bottom of the acrylic container are subject to different concentration levels during the filling process. As the radar TOF measurement is conducted in between the reference sensor positions the MOX results can be considered minimum and maximum boundaries of the expected ground-truth. Additional sensors acquire pressure, temperature and humidity inside the box. These results are used to adjust the permittivity values given in the literature for standardized conditions of 1atm pressure and 20°C to the environmental conditions present during the measurement [15,16].

### 2.6. Signal Processing

After transmitting the sampled time signal $x\left[n\right]$ to the host PC the complex frequency spectrum $X\left(f\right)$ is calculated using linear processing [28]. The signal phase
(20)$$\varphi =\arg\left(X\left(f_{\max}\right)\right)$$
is evaluated at the main peak $\left|X\left(f_{\max}\right)\right|=\max\left(\left|X\left(f\right)\right|\right)$. The phase difference Δφ to the reference measurement is calculated and inserted into (Equation 5) to obtain the relative TOF $\Delta \tau_{d}$ canceling out the static offsets $\tau_{\mathrm{ext}}$ caused by the setup. Before permittivity and volume fraction estimations are derived according to (Equation 8) and (Equation 9), additional time-variant errors are compensated as discussed in the following section.

### 2.7. Setup Error Compensation

The mechanical elements of the measurement setup are sensitive to changes of the environmental temperature causing a systematic time-variant error which cannot be eliminated by the initial calibration with a reference medium. It was shown in Reference [13], however, that for small temperature changes this deviation can be neglected for most parts of the structure except for the brass waveguide due to its length and low thermal capacity. Measuring both forward and backward transmission of the same propagation path through the MUT allows for compensation of this remaining error. In the following, indices 21 and 12 denote the direction of the propagation path, for example, $x_{21}\left[n\right]$ indicates the sampled time signal received at sensor 2 due to a FMCW chirp transmitted by sensor 1 (as defined in Figure 4) and vice versa.

As the waveguide link is circumventing the gas container the time-variant LO synchronization delay $\tau_{\mathrm{LO}}$ is significantly longer than the TOF through the MUT $\tau_{d}$. Therefore, the TX chirp signal of the master sensor (TX${}_{1}$ in Figure 6) arrives at the slave (RX${}_{2}$) first followed by the LO signal (LO${}_{2}$) resulting in a negative BB frequency
(21)$$\Delta f_{21}=\frac{B}{T}\cdot \left(\tau_{d}-\tau_{\mathrm{LO}}\right)\;.$$

In contrast, the TX signal of the slave MMIC (TX${}_{2}$) is sent with delay $\tau_{\mathrm{LO}}$ resulting in a BB frequency
(22)$$\Delta f_{12}=\frac{B}{T}\cdot \left(\tau_{d}+\tau_{\mathrm{LO}}\right)$$
when received at the master (RX${}_{1}$). A similar effect is encountered in cooperative radar systems [29]. As the radar transceivers used in this work output only real-valued time samples, all frequency components in the negative half-plane get mirrored into the positive half-plane, thus
(23)$$\Delta f_{21}^{\prime }=-\Delta f_{21}=\frac{B}{T}\cdot \left(-\tau_{d}+\tau_{\mathrm{LO}}\right)\;.$$

Combining the results of both forward (Equation 21) and mirrored backward path (Equation 23) the data can be split into a differential-mode component
(24)$$\frac{\Delta f_{12}-\Delta f_{21}^{\prime }}{2}=\frac{B}{T}\cdot \tau_{d}$$
canceling out the most prominent setup error and leaving only the measurement parameter of interest $\tau_{d}$, as well as a common-mode component
(25)$$\frac{\Delta f_{12}+\Delta f_{21}^{\prime }}{2}=\frac{B}{T}\cdot \tau_{\mathrm{LO}}$$
containing the time-variant LO delay. A calculation of the permittivity based on the differential component of forward and reverse path impacts the accuracy bounds derived in Section 2.2. According to the rules of uncertainty propagation, the variance of (Equation 24) is given by
(26)$$\mathrm{var}\left\{\frac{\Delta f_{12}-\Delta f_{21}^{\prime }}{2}\right\}=\frac{1}{4}\left(\mathrm{var}\left\{\Delta f_{12}\right\}+\mathrm{var}\left\{\Delta f_{21}^{\prime }\right\}\right)\approx \frac{1}{2}\mathrm{var}\left\{\Delta f\right\}\;.$$
for independent measurements. Although forward and backward path are highly correlated, the temperature-dependent deviations as well as changes due to the MUT are systematic and therefore have no influence on the signals’ variance.

For ease of understanding, the proposed error compensation method was derived in frequency regime, while the signal processing described in Section 2.6 is based on phase estimation. As the expected relative TOF is extremely small for most common gases, the phase does not exceed the unambiguous range of ±π and equivalent considerations can be derived in phase domain according to (Equation 20). Thus, it follows from (Equation 16) and (Equation 26) for the theoretical accuracy limit of a permittivity measurement:(27)$$\mathrm{var}\left\{{\hat{\varepsilon }}_{d,\mathrm{diff}}\right\}\geq \frac{1}{2}\cdot \mathrm{var}\left\{\Delta {\hat{\varepsilon }}_{r}\right\}={\left(\frac{4\;c_{0}}{2\pi }\right)}^{2}\cdot \frac{\varepsilon_{r,\mathrm{ref}}}{d_{m}^{2}}\cdot \frac{1}{\mathrm{SNR}\cdot f_{0}^{2}\cdot N}\;.$$

## 3. Results

### 3.1. Phase Difference Measurement

Multiple measurements were conducted with He and CO${}_{2}$ as inclusion gases in air to evaluate the performance of the distributed radar sensor prototype setup. The results were compared to both the theoretical accuracy limits and reference values obtained with a commercial VNA and MOX sensors as described in Section 2.5. Prior to each measurement the setup is calibrated with an air-filled gas container to eliminate static errors caused by external signal paths. Afterwards, the MUT gas flow is activated and the acrylic box is filled slowly to minimize turbulence. A constant flow rate of 10 L/min is maintained until the inclusion gas volume fraction $\zeta_{i}$ starts to settle at a minimum of 90% as measured by the MOX concentration sensors. TOF measurements are taken approximately every 12 s with the distributed low-cost radar and every 56 s with the commercial VNA during this filling process of about 10–15 min duration. The phase difference of He and CO${}_{2}$ to the air reference obtained by the synchronized radar sensors during the filling process is displayed in Figure 7 both before and after compensation of systematic time-variant errors. Minimum and maximum boundaries are calculated from MOX gas concentration results.

### 3.2. Gas Concentration Measurement

Figure 8a–d compare the concentration level of the inclusion gas calculated from TOF measurements with the two synchronized radar sensors to the VNA reference. Due to the dual allocation of antennas TX${}_{1}$ and TX${}_{2}$ in the setup configurations for prototype hardware and VNA, a simultaneous measurement is not possible and the data of both configurations was obtained in consecutive runs causing slight deviations in the filling process. Additionally, minimum and maximum boundaries obtained by the MOX sensors are shown for comparison.

### 3.3. Variance

The fixed mechanical dimensions of the measurement setup as well as the limited frequency range of the MMICs used in the hardware prototype heavily restrict the practical evaluation of the system’s accuracy. Furthermore, the small leakages occurring at the removable lid and the cable bushings of the gas container do not allow the inclusion gas concentration to be stable over a longer time frame. Therefore, a sufficient number of measurement repetitions to determine the sensor performance can only be achieved with an air-filled observation area. Figure 9 depicts the permittivity calculated from forward and backward path as well as the result of the error compensation method over 90 consecutive runs with an air-filled gas container. MSEs for phase and permittivity were determined to $\mathrm{MSE}\left\{\Delta \hat{\varphi }\right\}=1.2\times 10^{-2}\;$deg and $\mathrm{MSE}\left\{{\hat{\varepsilon }}_{r}\right\}=1.2\times 10^{-10}$ for the single path measurements and to $\mathrm{MSE}\left\{\Delta {\hat{\varphi }}_{\mathrm{diff}}\right\}=2.5\times 10^{-4}\;$deg and $\mathrm{MSE}\left\{{\hat{\varepsilon }}_{r,\mathrm{diff}}\right\}=2.7\times 10^{-11}$ for the differential component, respectively. A relative phase of φ=0deg to the reference measurement and the literature value of the dielectric constant for air $\varepsilon_{r}=1.000536$ were used as true values in the calculations.

To compare the achieved MSEs against the theoretical limits, the SNR present in the measurement data is estimated using distant parts of the associated range spectra (shown for the forward path in Figure 10), where no multi-path reflections or other influences of the setup’s surroundings are present. The acquired value of 74 dB is corrected for averaging and FFT processing gain as well as windowing losses of 31 dB in total, resulting in an equivalent SNR=43dB in the sampled BB time signal. Consequently, the analytical CRLBs for phase and permittivity after setup error compensation calculate to $\mathrm{var}\left\{\Delta {\hat{\varphi }}_{\mathrm{diff}}\right\}=9.6\times 10^{-5}\;\deg$ and $\mathrm{var}\left\{{\hat{\varepsilon }}_{r,\mathrm{diff}}\right\}=1.1\times 10^{-11}$, respectively.

## 4. Discussion

Based on the comparison of the CRLBs of frequency- and phase-based TOF measurements in (Equation 12), phase estimation was chosen as basic measurement principle. For the prototype radar system’s parameters (Table 1) the theoretically achievable accuracy of a phase-based TOF shows an increase of 33 dB over the frequency estimation approach. With an exemplary SNR=43dB present in the measurement data the lower bound for the permittivity’s standard deviation calculated from frequency $\sigma_{f}=\sqrt{\mathrm{var}{\left\{\varepsilon_{r}\right\}}_{f}}\approx 2.5\times 10^{-4}$ is in the same order of magnitude as $\varepsilon_{r}-1$ for common gases and is thus not sufficiently accurate for most applications. On the other hand, a phase-based measurement can achieve a standard deviation up to $\sigma_{\varphi }=\sqrt{\mathrm{var}{\left\{\varepsilon_{r}\right\}}_{\varphi }}\approx 3.1\times 10^{-6}$, which is in the order of the precision usually found for the dielectric constant of gases in the literature. Despite the challenging implementation using low-cost hardware without temperature stabilization [20], the substantial gain in accuracy justifies the additional effort, for example, for time-variant error compensation. Due to the low differences of the dielectric constant of most common gases the unambiguous range of ±π is rarely exceeded in the targeted industrial applications.

The CRLB for permittivity (Equation 14) outlines various ways to improve the measurement accuracy:The radar sensor should be operated at the upper at end of the RF frequency range to maximize the FMCW chirp start frequency $f_{0}$.Calibration should be performed with a low-permittivity material, ideally vacuum.The mechanical dimensions of the measurement setup should be chosen to maximize the length of the propagation path inside the MUT $d_{m}$.

Equation (Equation 18) shows that the achievable performance for gas concentration measurements highly depends on the application, especially the gaseous substances involved. While it is possible to determine the volume fraction with high precision for gases with sufficient difference in permittivity, the variance increases significantly once environmental and inclusion gas are too similar in their dielectric properties. All theoretical CRLB results in Section 2.2 were confirmed by extensive Monte-Carlo simulations which show very good agreement to the analytical values.

Due to restrictions imposed by the implementation of the proposed system concept, for example, fixed mechanical dimensions and leakage through small openings in the gas container, the practical verification of the derived CRLBs is limited to TOF measurements with an air-filled container. Although only 90 consecutive runs were performed and the data is not sufficient to compute the MSEs of relative phase and permittivity with a high confidence level it was shown that the MSEs are only by a factor of 2.6 higher than the theoretical performance limits. To achieve this, systematic time-variant deviations caused by changes in the environmental conditions had to be corrected in post-processing. The compensation method proposed in Section 2.7 combines data of identical forward and reverse paths through the MUT. In the implemented setup, however, both paths are not physically identical due to the bi-static arrangement of the antennas in the configuration for the low-cost radar prototypes (see Figure 4). As the flow rate of the inclusion gas is kept low to minimize turbulence and the offsets of TX and RX antennas are small compared to the overall dimensions of the observation area the radar system can be approximated as mono-static. The high effectiveness of the compensation method confirms the validity of this assumption.

The performance of the transmission setup with two distributed, synchronized radar sensors was additionally compared to reference measurements with a commercial VNA and an estimate of the ground-truth obtained by MOX sensors. For both He and CO${}_{2}$ as inclusion gases inside air, the radar signal phase shows very good agreement after setup error compensation is applied, while the uncorrected signal deviates considerably from the estimated ground-truth. Volume fraction results demonstrate a performance of the low-cost prototype hardware similar to a commercial VNA at a fraction of cost, with the obtained concentration values located mostly within the expected min/max boundaries.

A factor of uncertainty when evaluating the measurement results is represented by the gases used in the experiments, especially the He. While the CO${}_{2}$ is specified to 99.9% purity, the He is only garantueed to be of minimum 90% purity by the supplier. Any contamination by other gases would alter the dielectric properties of the MUT, thus influencing the interpretation of parameters derived from the measured permittivity.

The proposed measurement method to determine the volume fraction of an inclusion gas inside an environmental gas is limited to binary gas mixtures. If multiple inclusion gases with unknown concentration levels are present simultaneously it is no longer possible to conclude their individual contribution to the effective permittivity $\varepsilon_{r,\mathrm{eff}}$. On the other hand, inclusion or environmental gases can consist of multiple gaseous substances as long as the composition is known and constant, for example, air. In this case, an effective permittivity is used to model the gas mixture as demonstrated in Section 3.

A remaining challenge is the detection of extremely small relative TOF values which occur for gases with similar dielectric properties or due to very short propagation paths through the MUT. Actively exploiting multi-path propagation and reflections inside the gas container will be investigated as a possible method to artificially increase $d_{m}$ wherever mechanical size restrictions apply. With the current implementation the waveguide synchronization link between master and slave sensor dominates the overall cost of the system. Therefore, alternative ways of synchronization employing only low-frequency signals are of interest to us. Although the temperature dependency of the measurement setup was reduced significantly by the proposed error compensation method, additional time-variant errors slowly increase over time and require re-calibration on a regular basis which might not be feasible in an industrial application. Ensuring long-term stability in harsh environments is another focus of future research.

## 5. Conclusions

In this work, the performance limits of two synchronized low-cost radar systems as alternative gas sensing solution were evaluated. First, the CRLBs for permittivity and concentration measurements were derived analytically. All results were validated by extensive Monte-Carlo simulations. Based on this, in the next step a TOF transmission sensor was implemented using fully-integrated FMCW radar MMICs operating in a master-slave configuration. He and CO${}_{2}$ as inclusion gases inside air were measured at varying volume fractions to test the proposed system concept experimentally. The resulting permittivity and concentration results show excellent agreement to reference values obtained with a commercial VNA and MOX gas sensors. Although determining the MSEs with high confidence level was not possible due to limitations of the measurement setup, the achieved accuracy was shown to be in the same order of magnitude as the theoretical performance limits. The proposed method for compensating time-variant setup errors proofed to be highly effective.

Future research will focus on variations of environmental parameters such as temperature or pressure to improve long-term stability. Additionally, alternative synchronization techniques will be investigated to further reduce the system cost and complexity.

## Figures and Tables

**Figure 1 sensors-19-03351-f001:**
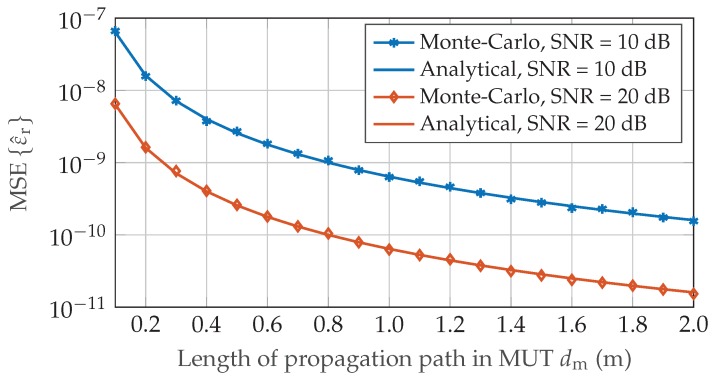
Monte-Carlo simulation of the mean square error $\mathrm{MSE}\left\{\varepsilon_{r}\right\}$ for N=4096, $f_{0}=77$ GHz and air reference $\varepsilon_{r,\mathrm{ref}}=1.000536$. The length of the propagation path in the medium under test (MUT) $d_{m}$ is varied in the range 0.1–2.0 m.

**Figure 2 sensors-19-03351-f002:**
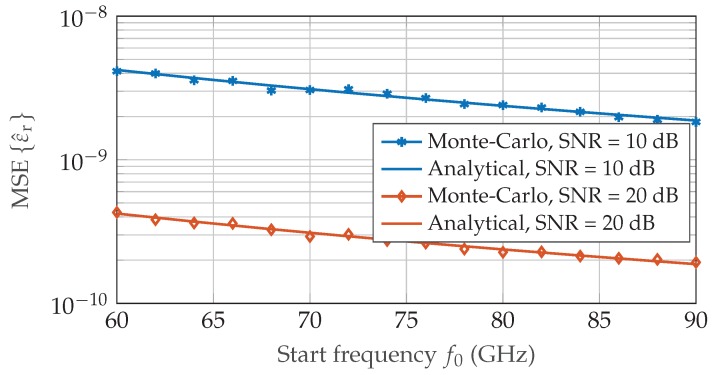
Monte-Carlo simulation of the mean square error $\mathrm{MSE}\left\{\varepsilon_{r}\right\}$ for N=4096, $d_{m}=0.5\;$m and air reference $\varepsilon_{r,\mathrm{ref}}=1.000536$. The start frequency of the frequency modulated continuous wave (FMCW) chirp signal $f_{0}$ is varied in the range 60–90 GHz.

**Figure 3 sensors-19-03351-f003:**
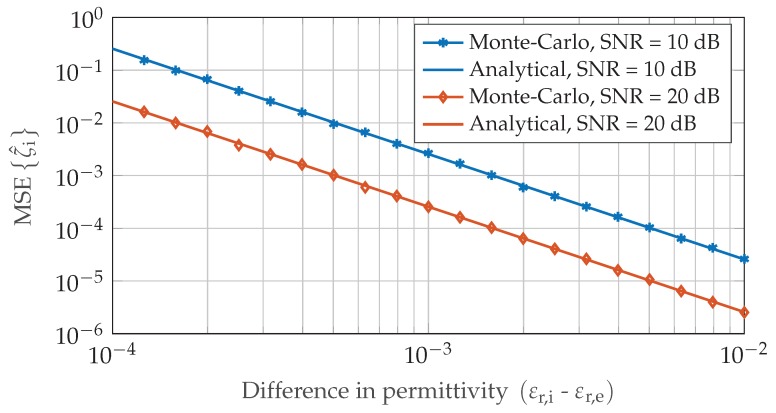
Monte-Carlo simulation of the mean square error $\mathrm{MSE}\left\{\zeta_{i}\right\}$ for N=4096, $f_{0}=77\;$ GHz, $d_{m}=0.5\;$ and air as reference and environmental gas with $\varepsilon_{r,\mathrm{ref}}=\varepsilon_{r,e}=1.000536$. The inclusion gas permittivity is sweeped to vary the permittivity difference $\varepsilon_{r,i}-\varepsilon_{r,e}$ in the range $10^{-4}$ to $10^{-2}$.

**Figure 4 sensors-19-03351-f004:**
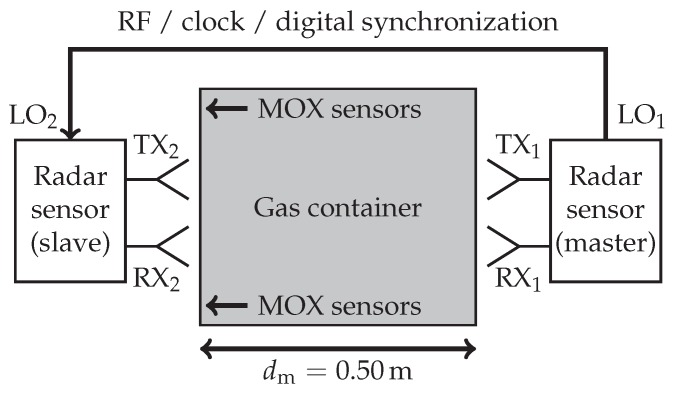
Schematic illustration of the measurement setup, showing the gas container, antennas and metal oxide (MOX) sensor mounting positions. Two synchronized, bi-static radar sensors are placed on opposing sides, transmitter and receiver of the master MMIC are connected to antennas TX${}_{1}$ and RX${}_{1}$, for the slave MMIC to TX${}_{2}$ and RX${}_{2}$, respectively. RF, clock and digital synchronization signals are forwarded from master to slave sensor via a waveguide and cables. When configured for the VNA reference measurement, the radar sensors are replaced by mono-static VNA ports connected to antennas TX${}_{1}$ and TX${}_{2}$, while antennas RX${}_{1}$ and RX${}_{2}$ as well as the synchronization link are not in use.

**Figure 5 sensors-19-03351-f005:**
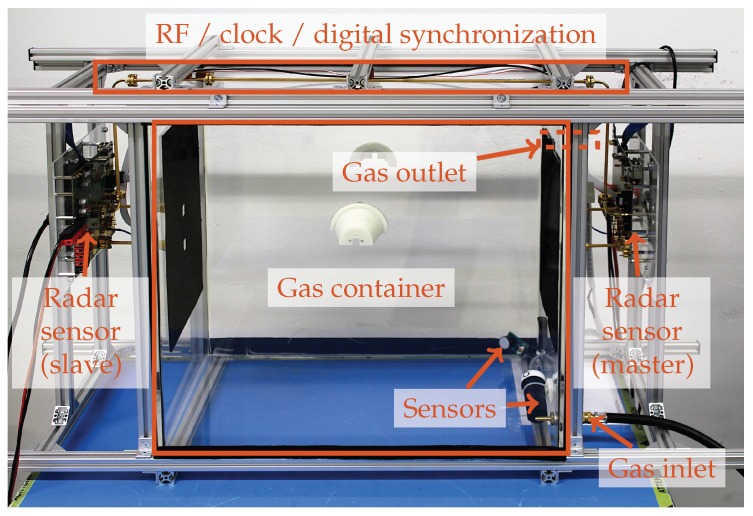
Photograph of the measurement setup with two synchronized low-cost radar sensors in master-slave operation mode. The synchronization link consisting of a WR-12 waveguide and multiple cables runs along the outside of the gas container. Multiple sensors for concentration, pressure, temperature and humidity are visible inside the box. The inclusion gas source is connected to the bottom inlet, the gas outlet on top is hidden by the mechanical support structure.

**Figure 6 sensors-19-03351-f006:**
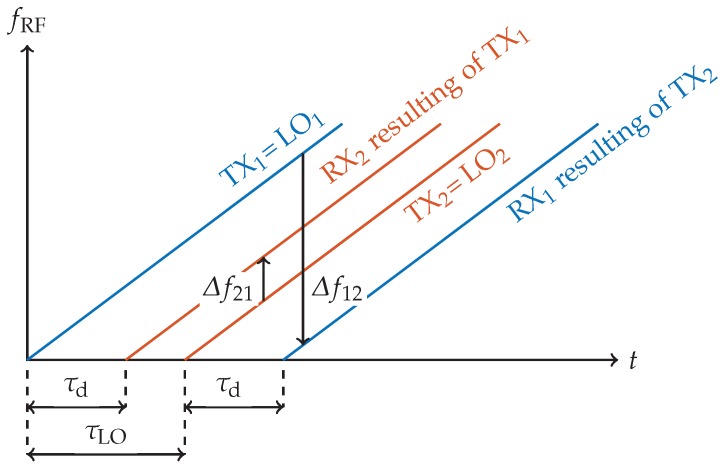
Schematic representation of the FMCW chirp signals transmitted and received by master and slave MMICs. Compared to the transmit signal of the master (TX${}_{1}$) the slave’s transmit signal (TX${}_{2}$) is delayed by the time-variant propagation time of the synchronization signal $\tau_{\mathrm{LO}}$. For both forward and backward transmission the TX chirps are received at the opposite sensor after TOF $\tau_{d}$ propagating along the direct path through the medium. The resulting BB frequencies $\Delta f_{12}$ and $\Delta f_{21}$ after down conversion of RX${}_{1}$ and RX${}_{2}$ with the respective LO signals LO${}_{1}$ and LO${}_{2}$ are depicted as arrows indicating the sign of the frequency difference.

**Figure 7 sensors-19-03351-f007:**
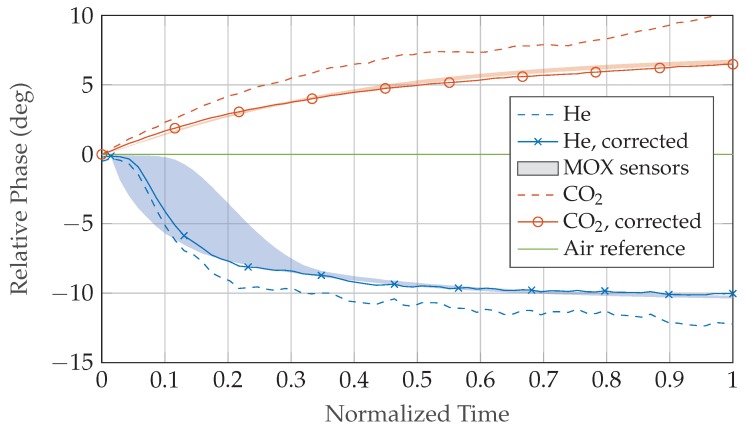
Phase difference of He and CO${}_{2}$ to the air reference evaluated at $\Delta f_{12}$ during the filling process. Data was acquired in a transmission measurement with two opposing synchronized radar sensors. Minimum and maximum boundaries are derived from MOX gas concentration results. After compensation of time-variant setup deviation is applied the differential-mode component shows good agreement to the minimum and maximum boundaries.

**Figure 8 sensors-19-03351-f008:**
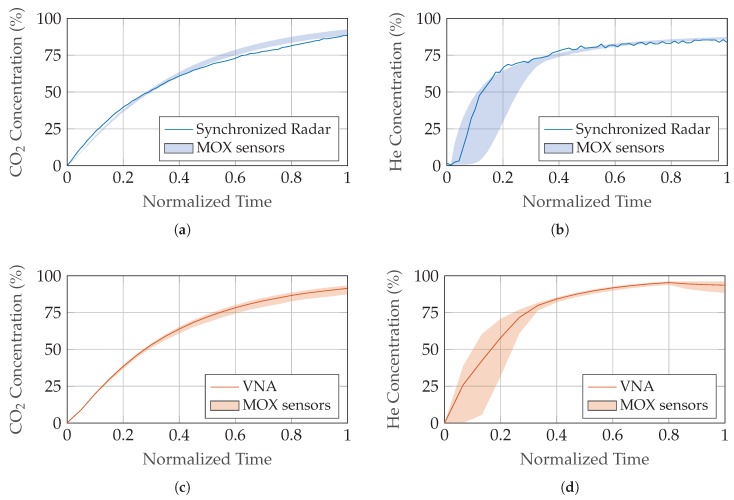
Increasing inclusion gas concentration during the filling process derived from relative TOF. Two measurement methods are shown in comparison to the estimated ground-truth boundaries obtained by MOX gas sensors. (**a**,**b**) Transmission measurement with two synchronized radar sensors; (**c**,**d**) S-parameter reference measurement with a commercial VNA. The results were generated in separate, consecutive runs, thus the filling process is not identical and the data is plotted separately. (**a**,**c**) CO${}_{2}$; (**b**,**d**) He.

**Figure 9 sensors-19-03351-f009:**
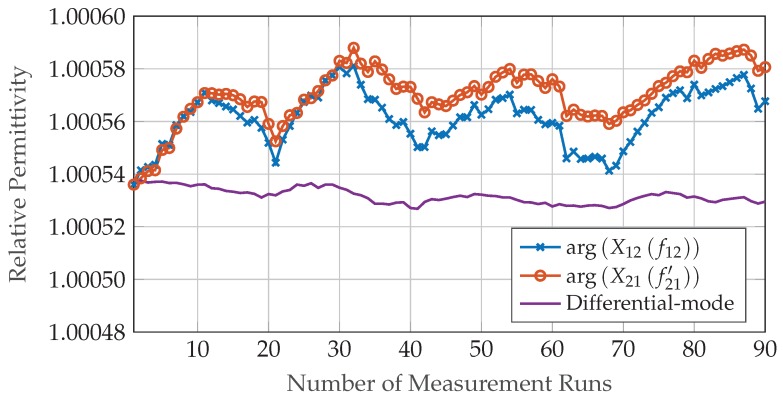
Relative permittivity of air filled gas container calculated from phase-based time-of-flight (TOF) over 90 consecutive measurement runs with two synchronized low-cost radar sensors. Forward and backward transmission path are shown separately as well as after time-variant error compensation.

**Figure 10 sensors-19-03351-f010:**
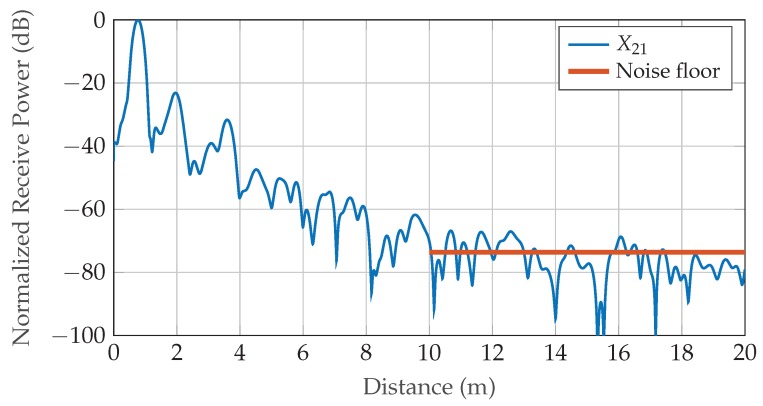
Normalized range spectrum of forward path TOF measurement averaged over 90 consecutive runs with air-filled gas container. Noise floor is estimated using power level present in range 10–20 m, where influences of the setup’s surroundings and multi-path reflections are neglectable.

**Table 1 sensors-19-03351-t001:** Radar sensor configuration used to obtain the results in Section 3.

Parameter	Value
Chirp bandwidth *B*	1.82GHz
Sweep time *T*	51μs
Start frequency $f_{0}$	78.82GHz
Sampling frequency $f_{s}$	5MHz

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
