# Peer review of "Accuracy Bounds and Measurements of a Contactless Permittivity Sensor for Gases Using Synchronized Low-Cost mm-Wave Frequency Modulated Continuous Wave Radar Transceivers"

_sensors, 2019, doi:10.3390/s19153351_

Round 1

Reviewer 1 Report

This paper proposed a novel idea to measure gas concentration by using bi-static FMCW radar sensors. The quality of this manuscript is pretty good. The authors provided sufficient theoretical analysis, and detailed descriptions of system setup and experiment.

There is one minor issue for the current version. Section 3.1 seems not very correlated to the other parts of the paper. This section was used to verify the correctness of CRLB by simulation. I think it may be better to put it below section 2.2, or combine it with section 2.2.

Author Response

Thank you for your positive review and your suggestion to improve the quality of the paper.

I can see that Section 3.1 feels displaced along the other results presented in Section 3. Therefore, I combined it with Section 2.2 where the CRLBs are theoretically derived. In this way, Section 3 focuses on experimental results only and the logical flow is improved.

Please find attached the revised version of the paper with the changes highlighted.

Reviewer 2 Report

This paper presents a radar-based method for permittivity measurement,

which includes detailed theoretical derivation and experiments.

Besides, this paper is well organized and written.

I have some questions listed as follows:

1. Since the gas has various components such as He and CO2, how to judge its concentration when there is a permittivity change? Please add a paragraph to describe the application scenario and limits of this method.

2. Since permittivity is frequency-dependent, would this phenomenon degrade the measurement accuracy within the operating bandwidth of 1.82 GHz.

3. Please add a table to compare with other radar-based studies.

Author Response

Thank you for your positive review and your suggestions to improve the quality of the paper.

1. The described method is applicable only to binary gas mixtures of one inclusion gas inside one environmental gas. As the measurement allows to determine the effective permittivity, the volume fraction of the inclusion gas can be calculated according to Equation 9. If multiple inclusion gases with unknown concentration levels are present it is no longer possible to solve for the individual volume fractions.

Therefore, in our experiment we demonstrated the method with He and CO2 as inclusion gases inside air as environmental gas in two separate measurements. In this way, only one inclusion gas at a time was present with unknown concentration level.

Inclusion gas or environmental gas can be gas mixtures as it is the case with air in our experiment. In this case, the composition of the gas mixture has to be constant and known for the gas mixture to be modeled with an effective permittivity.

We extended Section 4 to clarify the limits of the proposed method.

2. The dielectric constant of most gaseous substances, especially natural gases such as He and CO2, shows a very small dependency on frequency above 1 MHz. See for example [1] where the dielectric constant is found to be nearly identical in the radio frequency, microwave and optical range for multiple common gases including He and CO2. Within the operating frequency of 77–79 GHz the permittivity value of the gases involved in our experiment can therefore be considered constant without affecting the measurement accuracy.

We added a statement to Section 2.3 to include this information.

3.  Radar-based permittivity measurements have been studied since many years but concentrate on solids or fluids as materials under test. Only recently permittivity measurements for gaseous substances became feasible due to advances in high-resolution radar. As the radar system’s requirements on accuracy vary significantly for solid and gaseous materials, a direct comparison of the results is difficult. The most relevant scientific publication on radar-based permittivity determination of gas is [2], which is cited as reference in the introduction.

To the best of the authors’ knowledge there is, unfortunately, not enough relevant data available for an informative comparison in table format.

Please find attached the revised version of the paper with the changes highlighted.

[1] Maryott, A.A.; Buckley, F. Table of Dielectric Constants and Electric Dipole Moments of Substances in the Gaseous State. National Bureau of Standards Circular. National Bureau of Standards, 1953, Vol. 537.

[2] Baer, C.; Musch, T.; Jaeschke, T.; Pohl, N. Contactless determination of gas concentration and pressure based on a low jitter mmWave FMCW radar. Proc. IEEE Sensors Applications Symp. (SAS), 2014, pp. 11–14. doi:10.1109/SAS.2014.6798907.
